# Development of Proportional–Integrative–Derivative (PID) Optimized for the MicroElectric Discharge Machine Fabrication of Nano-Bismuth Colloid

**DOI:** 10.3390/mi11121065

**Published:** 2020-11-30

**Authors:** Kuo-Hsiung Tseng, Chaur-Yang Chang, Yagus Cahyadi, Meng-Yun Chung, Chin-Liang Hsieh

**Affiliations:** 1Department of Electrical Engineering, National Taipei University of Technology, Taipei 10608, Taiwan; cychang@ntut.edu.tw (C.-Y.C.); yaguscahyadi@gmail.com (Y.C.); alexmychung@gmail.com (M.-Y.C.); 2Management Department, Infosystem Technology, Hsinchu 30062, Taiwan; andre@infosystem.com.tw

**Keywords:** electrical spark discharge method, micro-EDM, PID, nano-bismuth colloid

## Abstract

Metal nanoparticles are typically prepared by using a chemical method, and a suspension is added to control the particle size and concentration of the nanoparticles. In this study, a micro-electric discharge machine (micro-EDM) was used to melt bismuth into nanoparticles, thus yielding a colloidal solution. No chemicals were added during the manufacturing process, and pure water was used as the medium. The colloid was assessed using an electrohydraulic system, and process parameters were adjusted for optimization; additionally, the discharge pulse wave was analyzed. The proposed preparation process is simple, fast, and cost-effective; moreover, the manufacturing process allows for mass production and reduces environmental pollution. Experimental results revealed that the nano-bismuth (nano-bi) colloidal solution was successfully prepared by the micro-EDM, and absorption peaks in the UV-vis spectrum were observed at 234 and 237 nm. Moreover, to optimize the proportional–integral–derivative (PID) control parameters to be used in the micro-EDM to prepare the nano-bi colloidal solution, this study derived a mathematical model of the micro-EDM. MATLAB was used to obtain the PID parameters. The discharge success rate (74.1876%) for the nano-bi colloidal solution prepared using our method was higher than that (46.9196%) obtained for a nano-bi colloidal solution prepared using an online adaptation method.

## 1. Introduction

The nanometer is a unit of length that is equivalent to one-billionth of a meter. In general, materials smaller than 100 nm are called nanograde materials, and the corresponding products are called nanoproducts. Because many metals have properties such as a low melting point, large surface, interfacial activity, and quantum effects at the nanometer scale [1], numerous nanometals have been developed and used in numerous applications; for example, they have been used as metal catalysts [2], absorption materials [3], and biomarkers [4] and have been employed in biomedical [5] and national defense [6] applications.

Bismuth is a chemical element with atomic number 83. Bismuth has several remarkable properties, such as a high specific gravity, low melting point, expansion and contraction under cold conditions during solidification, and nontoxic and noncarcinogenic effects on earthworms; therefore, it has been extensively used in numerous applications [7]. Moreover, bismuth is widely used in the metallurgical, chemical, electronics, aerospace, and pharmaceutical industries [8]. Some bismuth compounds are used as drugs for treating gastric dysfunction, such as diarrhea, colitis, and peptic ulcers. Medications containing bismuth are used as astringents; they exhibit antidiarrhea and antiseptic effects and can adsorb toxins, bacteria, and viruses. Recently, many methods for preparing nanometals have been developed; nanometals for use in commercial applications are mainly prepared through chemical methods, which allow them to be mass-produced [9,10]. However, nanometal colloids prepared through physical methods have many advantages for back-end applications. Several studies have been conducted on the preparation of nanometals through physical methods. The electrical spark discharge method (ESDM) [11,12,13] has been used to prepare nanometal particles immediately at room temperature, with no chemical byproducts generated during the process. This method is simple, fast, and environmentally friendly [14,15]. In the ESDM, a spark is generated through electric discharge to cut a workpiece into the desired shape. The dielectric liquid separates the two electrodes, and a voltage is applied to produce a periodically varying current discharge to process the material. One of the electrodes is called a tool electrode, or a pole tip, and the other electrode is called a workpiece electrode, or simply a workpiece. During the process, the tool and workpiece electrodes do not actually come into contact.

When the potential difference between the two electrodes increases, the electric field between them increases until the electric field strength is higher than the dielectric strength. At this point, dielectric breakdown occurs, and current flows between the two electrodes. The capacitor collapses (see the breakdown voltage), removing some of the electrode material. When the flow of current is stopped, a new dielectric flows through the electric field between the electrodes, eliminating solid particles. The dielectric then acts as an insulator again. After the flow of current is resumed, the potential difference between the two electrodes returns to the dielectric collapse; therefore, a new dielectric breakdown cycle can be repeated. The spark discharge process is usually followed by vulcanization, luminescence, and electromagnetic radiation [16]. It is a rather complex microscopic and transient physical process. The melting of a single discharge is divided into small stages that are independent and related to each other. As presented in Figure 1a, a polarized bridge is created when the electric field strength reaches a certain level. At this point, the insulating liquid molecules are polarized, forming a low-resistance channel between the two electrodes. When the electric field strength is above 106 V/cm, the anode begins to emit electrons. Subsequently, as illustrated in Figure 1b, electrons are emitted by the cathode under the action of the electric field; the electrons are accelerated and strike the molecules or atoms in the space between the two poles with intense force to ionize them. An ionized avalanche is observed from the cathode to the anode under the guidance of a low-resistance channel or a conductive bridge. Figure 1c shows that at the beginning of discharge, the discharge channel expands at a speed of several tens of meters per second. However, because of the magnetic compression effect caused by the increase in current at the discharge channel and the thermal compression effect caused by the cooling of the dielectric liquid, the discharge path is narrow. Owing to the dual action of expansion and contraction, the discharge channel has a small cross-sectional area; this means that the temperature (approximately 104 °C) and pressure (instantaneous pressure can reach tens to hundreds of atmospheres) in the discharge channel are very high. The rapid expansion of the discharge channel and the rapid drop in pressure causes some of the metal to melt or vaporize; this leads to the formation and gradual expansion of a bubble. As presented in Figure 1d, when the bubble shrinks to a certain extent, it is shattered, and the discharge melting process ends. Metal particles are suspended in the space between the electrodes and the liquid.

## 2. Materials and Methods

### 2.1. Micro Electric Discharge Machine

Figure 2 illustrates the design of a micro-electric discharge machine (micro-EDM) [17,18]. The vertical electrode configuration, which is usually observed in a traditional large-scale EDM, is replaced with a horizontal electrode configuration; this enables favorable utilization of the electrode material. A collet is used to clamp the electrode. One of the electrodes is placed in the small hole of a jig. The other electrode is fixed by a jig and connected to the discharge circuit. Constant weight is applied to the electrode to fix it on the left-hand side of the discharge circuit. The electrode on the right-hand side is fixed to a roller bearing slider. A DC motor (Hitachi D04A191E, Tokyo, Japan) controls the position of the slide on the slide rail; thus, the gap between the two electrodes is controlled. The aforementioned fixed chuck device and fixed electrode device are both 3D printed; therefore, the manufacturing cost is much lower than that of devices in an industrial-grade EDM. The magnetic stirrer under the container is composed of a permanent magnet fixed to a fan; when the fan rotates, the permanent magnet drives the stirring bar at the bottom of the container to rotate. The rotation of the magnet induces a stirring effect, allowing the metal nanoparticles in the container to uniformly disperse in the solvent. For convenience, a lifting table is placed under the magnetic stirrer. The lifting platform is raised during the process to enable the immersion of the metal electrodes in the dielectric liquid; the lifting platform is lowered after the completion of the process.

#### 2.1.1. Discharge Circuit

According to the discharge-type circuit design of the micro-EDM, a transistor is selected to provide the pulse voltage required for the discharge of the electrode, and the system is discharged according to the process parameters of the computer electronic control terminal [19]. Figure 3a shows the discharge circuit diagram. The system consists of a 100 V DC voltage source, a current limiting resistor RS, a transistor IRF740, an optical coupling (6N137) protection circuit, a current signal resistor R1, and voltage divider signal resistors R2 and R3. RS is used to limit the maximum value of the discharge current. IRF740 is used as a switch in the discharge circuit. IRF740 quickly turns on or turns off the input 100 V DC voltage and converts it into a high-frequency pulse signal; this signal acts as the electrical pulse power source of the electrode in the micro-EDM. The circuit design method of the transistor switching delay time cannot be too long; the switching extension time of IRF740 is within 51 ns, and therefore, it meets the requirement of fast switching for electrode discharge. The gate stage controls the conduction and breaking of the transistor. After the pulse width modulation (PWM) signal is received from the computer, the voltage and current are boosted by the 74LS07 buffer. Through the isolation of the optical coupling (6N137), the gate of the transistor is turned on, and a 100-V DC discharge pulse wave is output at the electrode. The discharge circuit is pulse-modulated according to the computer’s duty cycle. R1 is connected in series with the electrode terminal and has a resistance value of 1 Ω; it provides the discharge current signal. Therefore, the voltage drops when the discharge current signal flows through the electrode. Moreover, R2 and R3 are connected in parallel with the electrode terminal, and their resistance values are 1 and 19 kΩ, respectively. The discharge voltage is then attenuated by a factor of 20; the signal is then transmitted to other circuits for feedback control.

#### 2.1.2. Motor Control and Feedback Circuit

In this circuit, the signal of the discharge voltage between the electrodes is feedback for motor control. First, the voltage between the electrodes is sampled through a differential amplifier; the voltage signal is converted into an analog signal through an integrating capacitor. Finally, the analog signal is sent to the computer control terminal through the isolation circuit. The computer uses the signal to control the DC motor operation [20]. Figure 3b shows the flow of the current signal. The computer outputs motor control commands based on the analog signal of the integrated voltage. After the voltage regulation and circuit isolation protection, the command signal is transmitted to the motor driver (ref) to drive the DC motor. The sliding track is adjusted (through forward or backward movement) to maintain the electrode discharge interval within an optimum range. The DC motor contains an optical encoder that returns information regarding the motor’s movement to the monitoring interface so that the system gap or displacement data in each process can be derived and stored.

#### 2.1.3. Logic Determination Circuit

This circuit is used to determine whether the discharge is successful. The system uses this signal to calculate the discharge success rate and the energy consumed by the electrodes during processing. The distance between the electrodes determines the success of the discharge process, and the voltage across the electrodes must be proportional [21]. The voltage signal (Vgap) and the current signal (Igap) from the discharge circuit are assessed logically. The circuit determines whether the system discharge is successful. Figure 3c shows the schematic of the logic determination circuit. The circuit mainly comprises a comparator and an AND logic gate. The circuit inputs are Vgap and Igap signals. Because the Vgap value between the electrodes drops, the Igap value increases when the system generates a discharge spark. From the synchronous time axis, both Vgap and Igap are observed simultaneously. The logic gate judgment loop is designed as the basis for determining the success rate. Finally, the discharge success signal (Vsuc) processed by the judgment loop is transmitted to the Connector Pin Terminal (CNT, the pin is located in RT-DAC4/peripheral component interconnect (PCI) card) pin of the computer. The software monitoring system calculates the success rate of the discharge and the energy consumed by the electrode according to the number of successful pulse discharges. The calculation result is instantly displayed on the computer screen. The discharge success rate provides an indication of whether the parameter settings are suitable.

#### 2.1.4. VisSim Monitoring System

The micro-EDM is designed with a PC as the main control core. VisSim software serves as the platform for system development. All input controls and signals from the hardware circuits are sent to the controllers, which are constructed using VisSim [22]. The screen of the controller is shown in Figure 4. The PC is connected to the RT-DAC4/PCI interface card for signal transmission and reception. The RT-DAC4/PCI interface card contains a field-programmable gate array; the chip has multiple sets of digital and analog I/O modules. Because the frequency of the pulse wave modulation is between 0 and 50 kHz in the ESDM, the quantized frequency is approximately several hundred thousand times per second.

Nanometal processing using a micro-EDM requires the processing of different signals: a transistor control PWM signal, discharge voltage feedback signal, current feedback signal, and motor motion control signal (ref). Moreover, motor displacement feedback value-encoder and the discharge successful quantized signal and discharge successful count (Vsuc). Therefore, the RT-DAC4/PCI interface of the main controller must be able to transmit and receive high-frequency digital and analog signals quickly, as well as optical encoding encoder reading and counter CNT accumulating function. The VisSim software interface enables the achievement of 1-MHz digital signal synchronous output control. Through this system, an operator can use the computer monitoring interface for nanometal colloid processing.

### 2.2. Micro-EDM Preparation Process Optimization

Before nano-bi colloids are prepared using a micro-EDM, the system’s operating process through system control theory must be first understood. The debugging method for the proportional–integral–derivative (PID) control parameters includes a manual adjustment method and computer software auxiliary method; the parameters of the micro-EDM can then be adjusted after debugging to obtain the ideal parameters for preparing a nano-bi colloidal solution.

#### 2.2.1. Micro-EDM System Control

The micro-EDM uses a hardware circuit for feedback. The discharge voltage signal is converted into an analog signal through capacitor integration and sent to the VisSim software for calculation. The analog signal is compared with the system-set command value to determine the control action of the motor. Figure 5 displays a schematic of the conversion of the discharge voltage signal. Because the discharge pulse wave is being processed, the voltage decreases with the discharge success; therefore, the analog signal after the integration floats depending on the state of the electric discharge machine. It can be divided into three types: system open circuit, discharge success, and electrode short circuit.

#### 2.2.2. Micro-EDM System Parameters

Figure 6 illustrates the components of the micro-EDM system circuit board and the corresponding parameters. The components include a PID, digital-to-analog converter (DAC), motor, lead screw, Vgap feedback block, and Vgain. Some circuits are used only for signal buffering and isolation and do not affect the gain of the system; these circuits are omitted. Moreover, the component for calculating the discharge success rate is part of the observation component and is not included in the control block diagram. V_ref_ is an input control command for the system; it is set to 2 V. The PID control block comprises the VisSim software; this is the core part of the entire control system. The PID block is controlled by the error result of V_ref_ and V_feedback_. The output of the PID block is transferred to the DAC and converted to a voltage signal to control the DC motor.

During the preparation of nano-bi colloids, if the distance between the two electrodes is very large, the two electrodes remain in the open state. The controller must operate the motor to bring the two electrodes closer [23]. If the distance between the two electrodes is very small, the two electrodes become connected, resulting in a short circuit. This would result in a discharge feedback analog signal value of zero; therefore, the two electrodes must be separated. If the distance is adequately controlled, the system can achieve a high discharge rate. As presented in Figure 5, when the electrode voltage is converted to an analog signal when discharge is successful, the voltage value remains within a certain interval, and the motor ensures that a small distance is maintained between the two electrodes. Through this control principle, the system sets a comparison command value to maintain the electrode feedback signal within the interval of successful discharge and controls the motor action according to the comparison result.

#### 2.2.3. PID Tuning Methods

The parameters of the PID controller can be tuned using various approaches; the most effective approach is to develop a program and tune the parameters according to their dynamic characteristics [24,25,26,27]. Manual tuning is inefficient, particularly for systems with a response time of more than a few minutes. The choice of tuning method depends on whether the control loop can be temporarily taken offline; it also depends on the system response time. Offline conditions (e.g., no load) are slightly different from actual operating conditions; the controller needs to consider only a theoretical situation and not actual usage to determine the control output. Online tuning refers to conditions during actual usage; the controller should consider actual usage to determine the control output. If the control loop can be taken offline, the most suitable approach to debugging is to provide the system with a step input, measure its output with respect to time, and use its response to determine the parameters. This study uses manual tuning to set up the PID. It is simple and can be executed online; moreover, no calculations are required. However, manual tuning yields poor results and requires highly experienced personnel [28]. A PID tuner can be used to easily adjust the PID control parameters; however, the system must be a linear time-invariant (LTI) model [29]. During the execution of the PID tuner, the first step is to establish the LTI model of the micro-EDM system. This method is consistent, and tuning can be performed either online or offline. Moreover, this method may include valve and sensor analysis; it enables simulation before the actual test and supports nonsteady-state debugging. On the basis of the specifications in Figure 6, this study can derive the transfer function of the DC motor Equation (1). *P_lead_* is the transfer function of the lead screw Equation (2), *K_Amp_* is the magnification of the DC motor drive Equation (3), and *Ps* is the open-loop transfer function of the micro-EDM Equation (4). Subsequently, the PID tuner implemented on MATLAB calculates the values of Kp, Ki, and Kd (the parameter of the PID) as well as the response of the micro-EDM.
(1)$$M\left(s\right)=\frac{\dot{\theta }\left(s\right)}{V\left(s\right)}=\frac{K}{\left(Js+b\right)\left(Ls+R\right)+K^{2}}\;\;\;\left[\frac{rad/sec}{V}\right]$$
where Js is the moment of inertia of the rotor (8.5229 × 10^−7^ kg-m^2^), b is the viscous friction constant of the motor (9.4235 × 10^−6^ N m s), K is the electromotive force constant (0.025 V/rad/s), K*t* is the motor torque constant (0.025 N m/A), R is the electrical resistance (7.247 Ω), and Ls is the electrical inductance (2.2338 mH).
(2)$$p_{lead}=\;\frac{1}{s}\frac{l}{2\pi }$$
(3)$$K_{Amp}=\frac{Vdrive}{Vinput}$$
where *V_drive_* is the operating voltage of the DC motor, and *V_input_* is the input control voltage.

(4)$$Ps=M\left(s\right)\times \mathrm{Plead}\times Kamp$$

### 2.3. Analytical Instruments for Nano-Bi Colloids

This study conducted an experiment by using three measuring instruments to analyze different characteristics and attributes: a Thermo Helios Omega ultraviolet (UV)–visible (vis) spectrophotometer [30,31], a Zetasizer Nano System, and a transmission electron microscope (TEM, TEM-JEOL, JEOL Ltd., Tokyo, Japan). The absorption of the nano-bi colloids was analyzed using the spectrophotometer as the concentration index of nanoparticles. In UV–vis, the start and stop wavelength are 350 and 600 nm under the scanning speed and interval of 240 nm/min and 1 nm. The zeta potential represents the suspension stability of particles in a fluid [32,33]. The Zetasizer nanosystem (Zetasizer, Zeta potential, Nano-ZS90, Malvern Panalytical Ltd., Worcestershire, UK) was used to measure the surface potential of the particles and the particle size distribution. Generally, if the diameter of particles is less than 100 nm and the absolute value of the surface potential is higher than 30 mV, the particles are referred to as nanoparticles. In Zetasizer, the light source is He-Ne laser (633 nm), the scattering angle to measure particle size is 90 degrees. The dispersant setting of the Zetasizer is water with 25 °C in temperature, 0.8872 mPa s in viscosity and 1.330 in refractive index. The DW purity is 7 μS/cm in conductivity and 25 ppm of the dissolved solids. The TEM was used to observe the features of the very small specimens [34]. In TEM, an accelerated beam of electrons is passed through a very thin specimen to enable a scientist to observe structural and morphological features. In TEM, the energy is as high as 200 kV, and the magnification is 40,000×. Furthermore, the EDX attached to TEM is used to get composition analysis. In this study, each sample was measured through five steps sequenced in incremental order; if the measurement in each revealed the sample to be a non-bismuth material, then the parameters defined for the sample were deemed erroneous. Thus, the measurements were continued until the five steps simultaneously confirmed the sample to be bismuth with typical characteristics of the element. In addition, dispersed bismuth oxide was used as a substrate for concentration analysis.

## 3. Results

In this study, nano-bi colloids were prepared with the parameters as showed in Table 1:

### 3.1. Manual PID Setting

To determine the PID parameters through manual adjustment, first set Ki and Kd to 0 and gradually increased Kp. Then, it allowed the motor to drive the lead screw and let the two electrodes conduct. Then checked whether discharge existed between the two electrodes and calculated the discharge success rate using VisSim to confirm whether the electrodes were discharged. If no discharge was observed, Kp was increased, and the Ton and Toff times were adjusted. When the electrode started to discharge, Ki was gradually increased; then observed whether the lead screw was accelerated or advanced. When Ki was increased, the lead screw moved without change. Kd was gradually increased and checked whether VisSim’s error was reduced and whether the discharge success rate increased until the appropriate error or discharge success rate was attained or could not be changed.

As displayed in Figure 7, when Kp = 0.5, Ki = 0.05, Kd = 0.05, and Ton-Toff of 30–300 us, the results were favorable, and the discharge success rate was 46.9196%. The experimental results demonstrated that when Ki exceeded a certain value, the lead screw continued to move forward and exceeded the limit switch, and the motor was powered off and stopped working; furthermore, a serious integral saturation effect occurred. When Ki was equal to Kd, the integral saturation effect was reduced, and the motor continued to work.

### 3.2. PID Tuner to Optimize PID Setting

Figure 8 shows the results obtained when used the PID tuner implemented on MATLAB to determine Kp, Ki, and Kd. First, set the tuner’s Type to PID, after which import the micro-EDM open-loop transfer function Ps Equation (4). Subsequently, a set of recommended values for Kp, Ki, and Kd were automatically determined by the PID tuner. Figure 8a shows the step response of Ps, and Figure 8b shows its performance and robustness. Figure 8 shows the PID values obtained by PID tuner: Kp = 3.622, Ki = 74.2724, and Kd = 0.018846. These values would not have been obtained with manual tuning. The values obtained here were restricted by the upper and lower limits of VisSim’s original design. The limit of VisSim was determined to be 0–5; before the PID set was applied, the limit was changed to 0–100. However, the experimental results revealed that the micro-EDM lead screw was powered off after touching the limit switch. This phenomenon was also observed in the manual adjustment method when Ki exceeded a certain value; the micro-EDM exhibited an integral saturation effect, which caused the lead screw to overshoot. When Kd was equal to Ki (i.e., Ki = Kd = 74.2724), the micro-EDM operated effectively.

To analyze the Kp, Ki, and Kd results recommended by PID tuner, this study plotted step responses, as presented in Figure 9. This study considered two groups of parameters: Group 1 (Kp = 3.622, Ki = 74.2724, Kd = 0.018846) consisted of parameters obtained using PID tuner, and Group 2 (Kp = 3.622, Ki = 74.2724, Kd = 74.2724) contained modified parameters. Figure 9a shows the step response when the sampling time (Tsample) was 1 ms. Curve ① represents the step response of the parameters in Group 1, and curve ② represents the step response of the parameters in Group 2. As observed in Figure 9a, the step response of the parameters in Group 1 reached a maximum value of 1.12, and the rise time and settling time were 0.00694 and 0.0651 s, respectively. The step response of the parameters in Group 2 reached a maximum value of 1.013, and the rise time and settling time were 0.001 and 0.004 ms, respectively. Group 2 had a better step response. Figure 9b shows the step response when the sampling time was changed to 0.1 ms. The step response of the parameters in Group 2 oscillated at the beginning, and the maximum amplitude was 1.8411. Although the parameters in Group 2 caused a considerable overshoot, the high-frequency oscillation was ignored by VisSim. Then, applying the parameter in Group 2 into micro-EDM. As shown in Figure 10, the success rate in the preparation of nano-bi using the micro-EDM reached 74.1876%. This indicates that the results of the MATLAB simulation analysis are consistent with the VisSim results, proving that the mathematical model of the micro-EDM is accurate and can be used to achieve high nano-bi preparation success rates.

### 3.3. Analysis of Nano-Bi

#### 3.3.1. Peak Absorption of Nano-Bi Using UV–Vis

Figure 11 shows the UV absorption peak of a nano-bi colloidal solution prepared using the micro-EDM. Three curves are shown in the figure. The UV-vis absorption peaks of nano-bi were observed at 234 and 237 nm. For small particles, the absorption peak was at 234 nm, and for large particles, the absorption peak was at 237 nm. Table 2 lists the results observed in Figure 11. Entry 1 duty cycle is 50–300 us, Kp = 3.622, Ki = 74.2472 and Kd = 74.2724. It has an absorbance peak of 0.702, wavelength 237 nm and size 96.42 nm. Entry 2 duty cycle is 50–300 us, Kp = 3.622, Ki = 74.2472 and Kd = 0.018846. It has an absorbance peak of 0.345, wavelength 2 of 34 nm and size 90.9 nm. Entry 3 duty cycle is 30–300 us, Kp = 0.5, Ki = 0.05 and Kd = 0.05. It has an absorbance peak of 0.321, a wavelength of 237 nm and a size of 126.7 nm.

#### 3.3.2. Analyses Using TEM

The TEM (JEOL JEM-2100F) image revealed that the nano-bi dispersions had a particle size of less than 50 nm when formed in DW. Figure 12 shows a macroscopic TEM image. Large black particles were observed, which must be the large particles of bismuth measured using the Zetasizer.

#### 3.3.3. Analyses Using Energy-Dispersive X-ray Spectroscopy

An energy-dispersive X-ray spectroscopy (EDX) instrument (JEOL JEM-2100F) was used to reveal the chemical composition of nano-bi. Figure 13 shows the EDX spectrum of nano-bi deposited on a copper carrier. Nano-bi peaks were observed at 1.8, 2.5, and 13.1 keV. Figure 13 indicates that the components of the nano-bi colloidal solution were relatively strong. Furthermore, the nano-bi colloidal solution contained other components, including carbon, oxygen, copper, and thorium. Because the nano-bi colloidal solution was prepared using deionized water and a bismuth metal wire and was sampled on a copper mesh, the presence of bismuth, carbon, oxygen, and copper was normal. However, the presence of thorium was rather unusual. This study suspect that thorium was present in the bismuth metal wire; however, further analysis is required.

## 4. Conclusions

In this study, a nano-bi colloidal solution was successfully prepared using the ESDM. Analyses conducted using a Zetasizer, UV-vis instrument, TEM, and EDX instrument proved that the products of the solution were of nanograde. Moreover, a mathematical model of a micro-EDM was successfully implemented on the basis of theoretical derivation results. This study derived the PID control parameters of the micro-EDM by using MATLAB. The conclusions of the study are outlined as follows:
In this study, a nano-bi colloid successfully prepared by using the micro-EDM with parameter Kp = 3.622, Ki = 74.2472 and Kd = 74.2472 and Ton-Toff 50–300 us in 2 min;Preparing nano-bi use micro-EDM with manual adjustments in Kp, Ki and Kd have been made previously, but whether the derived parameters are optimal is unclear. In this study, each block of the micro-EDM and transfer function was examined. Then use MATLAB to determine the PID parameters of the micro-EDM model. The results indicate that Kp, Ki and Kd values obtained using MATLAB were suitable for application to the micro-EDM and the discharge success rate 74.1876%;In general, nanomaterials in sputum prepared through a micro-EDM are nonpolluting. This method requires no additional chemical materials and used pure water as a medium;A TEM and EDX confirmed that the colloidal solution prepared in this study was indeed composed of nano-bi. UV absorption peaks of nano-bi were found at 234 and 237 nm.


## Figures and Tables

**Figure 1 micromachines-11-01065-f001:**
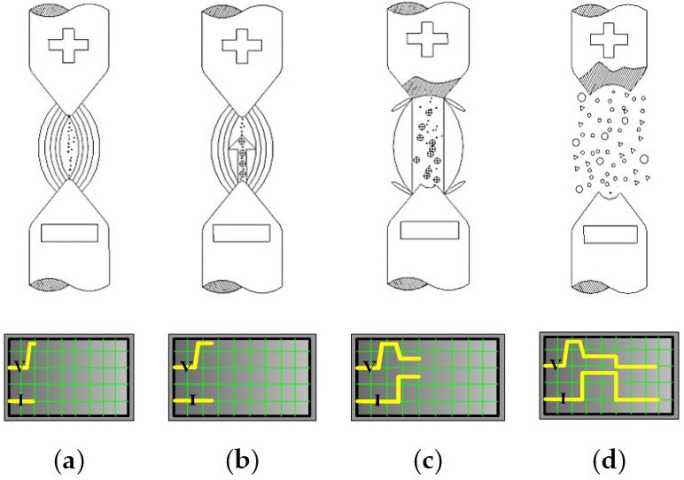
Steps involved in nanoparticle formation through micro-electric discharge machine (ESDM). (**a**) Molecular polarization and formation of a low-resistance channel. (**b**) Ionization and emission of electrons from the cathode. (**c**) Metal melt and vaporization. (**d**) Suspension of particles.

**Figure 2 micromachines-11-01065-f002:**
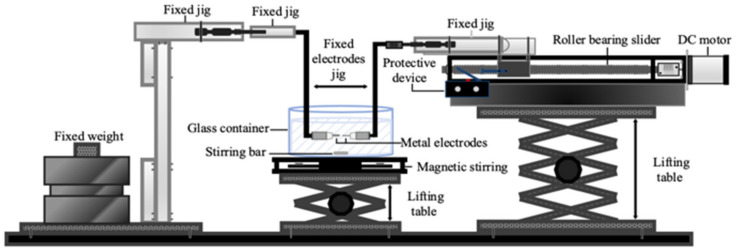
Design of micro-EDM.

**Figure 3 micromachines-11-01065-f003:**
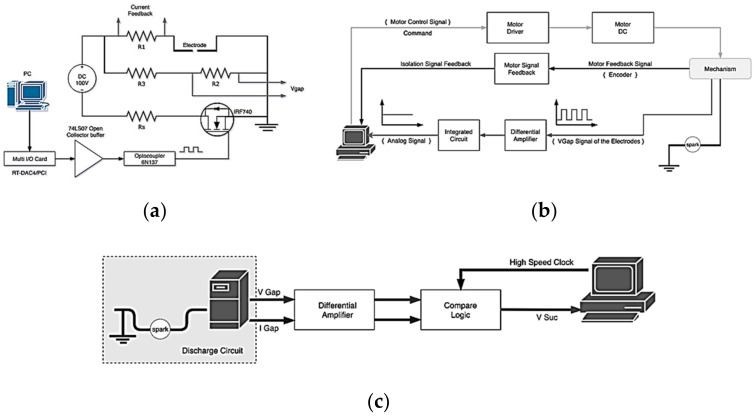
Circuit blocks of micro-EDM. (**a**) Discharge circuit, (**b**) motor control and signal feedback circuit, and (**c**) logic determination circuit.

**Figure 4 micromachines-11-01065-f004:**
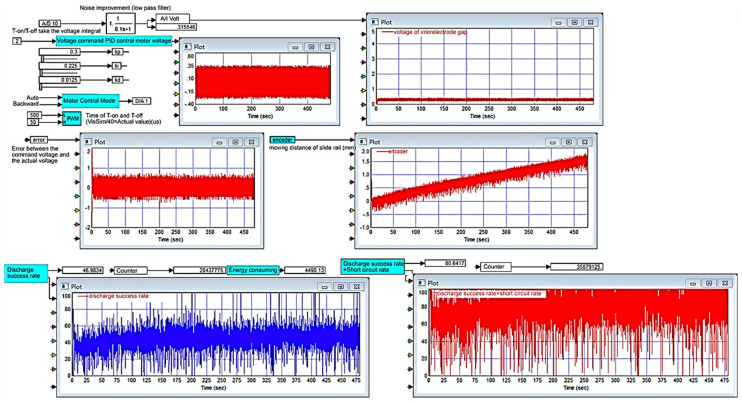
Monitoring interface developed by VisSim software.

**Figure 5 micromachines-11-01065-f005:**
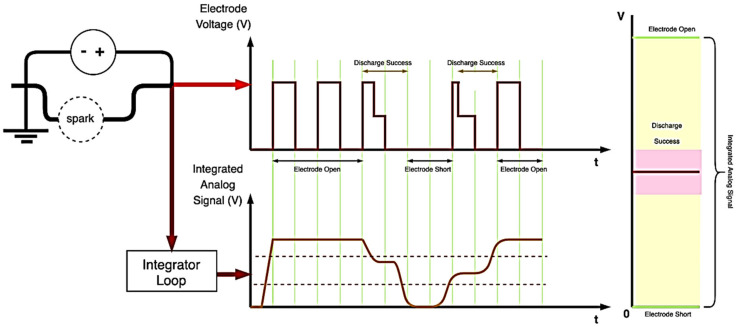
Schematic of discharge voltage signal conversion.

**Figure 6 micromachines-11-01065-f006:**
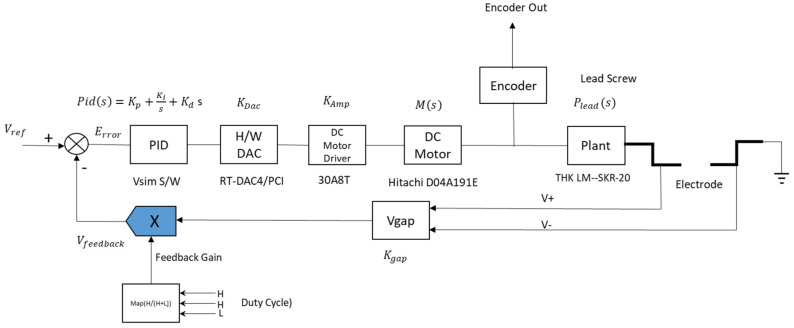
Micro-EDM system parameters.

**Figure 7 micromachines-11-01065-f007:**
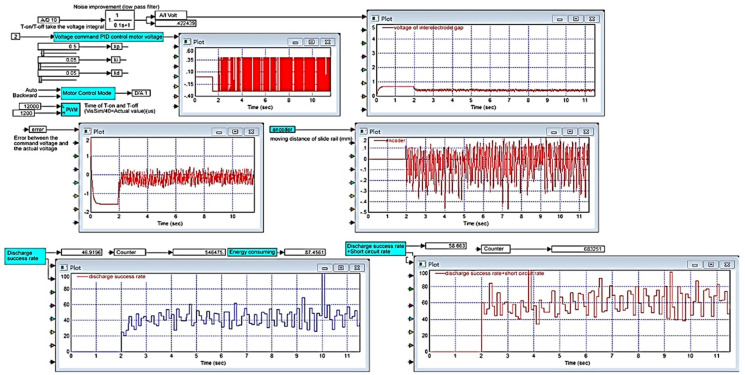
Kp = 0.5, Ki = 0.05, and Kd = 0.05. After Kd increases, error decreases, the discharge success rate increases and the system trends stabilize.

**Figure 8 micromachines-11-01065-f008:**
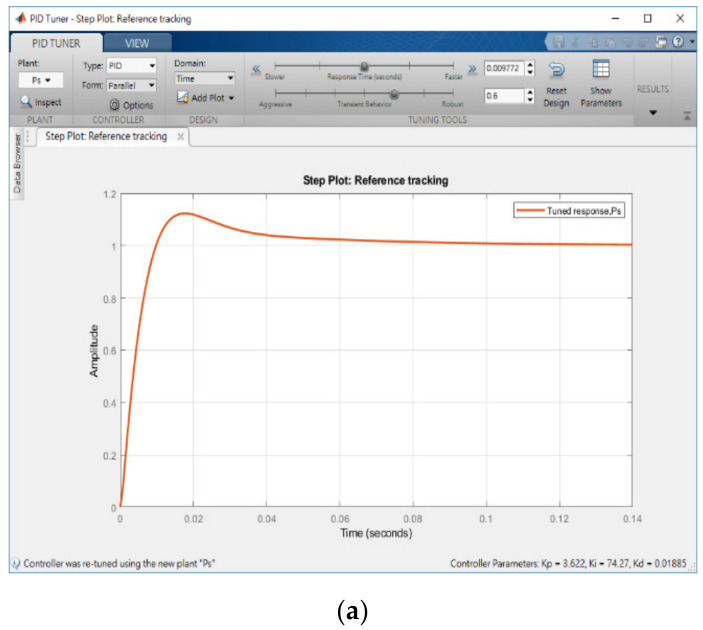
Proportional–integral–derivative (PID) tuner. (**a**) Step plot PID tuner and (**b**) PID value, performance, and robustness.

**Figure 9 micromachines-11-01065-f009:**
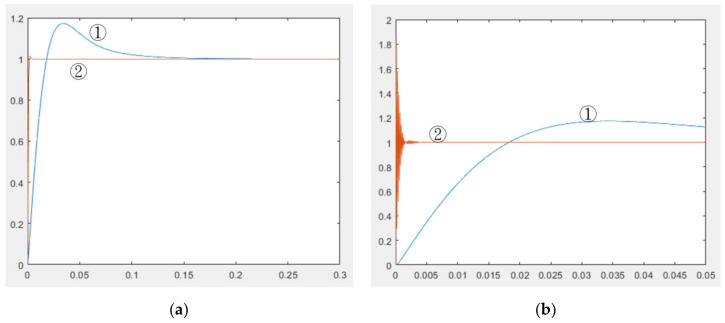
Step response (**a**) with a sampling time of 1 ms and (**b**) with a sampling time of 0.1 ms.

**Figure 10 micromachines-11-01065-f010:**
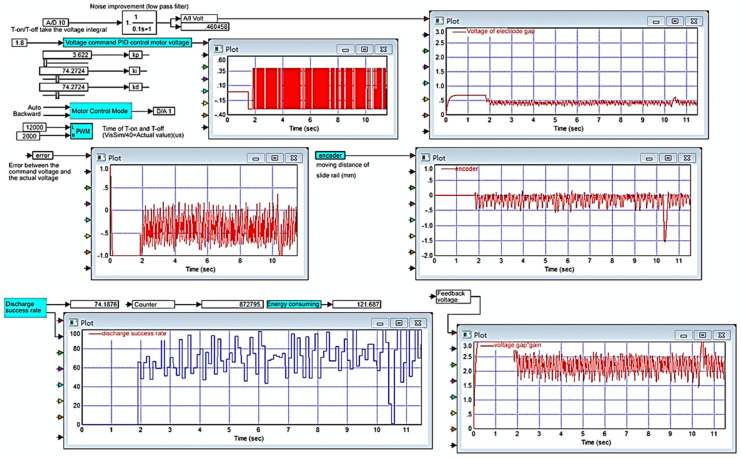
Process monitoring chart for preparing nano-bi. Kp = 3.622, Ki = 74.2724, and Kd = 74.2724.

**Figure 11 micromachines-11-01065-f011:**
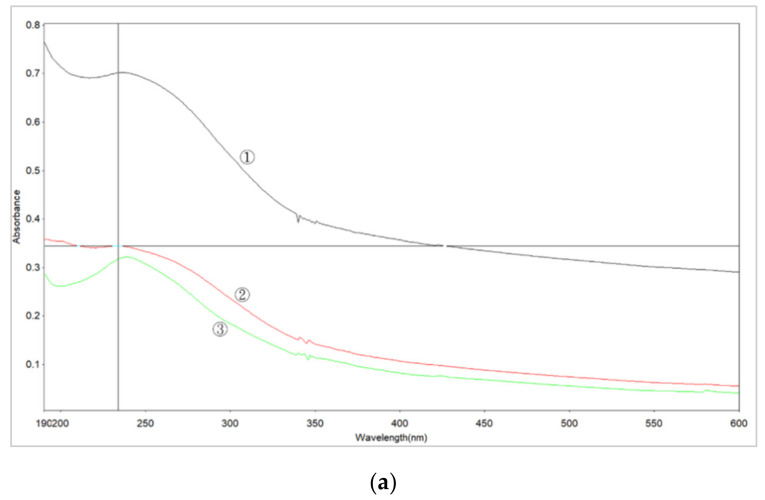
Absorption peak curves of a nano-bicolloidal solution prepared by micro-EDM. (**a**) Absorption peak curves (**b**) size (duty cycle is 50–300 us) and (**c**) zeta potential (duty cycle is 50–300 us).

**Figure 12 micromachines-11-01065-f012:**
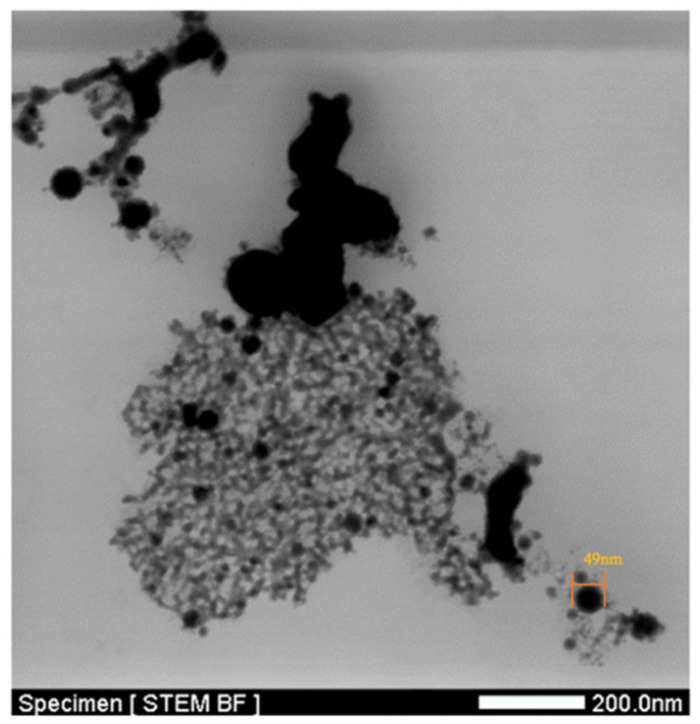
Macroscopic TEM image of bismuth nanoparticles.

**Figure 13 micromachines-11-01065-f013:**
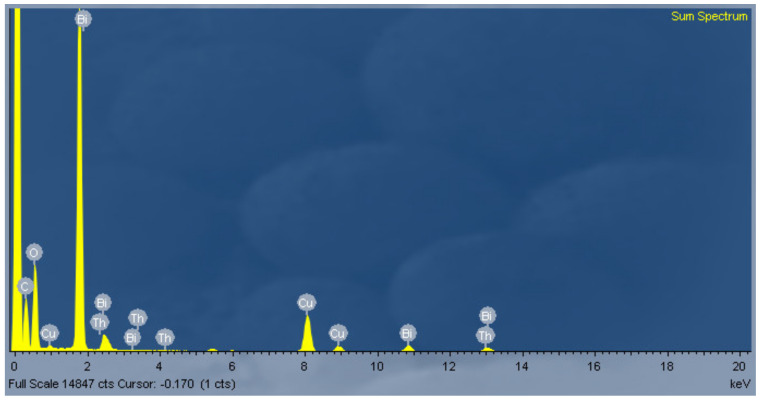
EDX spectrum of nano-bi.

**Table 1 micromachines-11-01065-t001:** Preparation parameters.

Experimental Parameter	Values
T_on_–T_off_	30–300, 50–300 us
Discharge time	2 min
Electrode	99.99% bismuth rods with diameter 3.175 mm and length 100 mm
Temperature	25 °C (room temperature)
Atmospheric pressure	1 atm
Dielectric fluid	200 mL deionized water
Ki, Kp, Kd	3.622, 74.2724, 0.018846; 3.622, 74.2724, 74.2724; 0.5, 0.05, 0.05

**Table 2 micromachines-11-01065-t002:** Parameters of micro-electric discharge machine (micro-EDM) and UV-vis absorption for nano-bi.

Entry	Duty Cycle (us-us)	Kp, Ki, Kd	Absorption Peak(À)	Wavelength (nm)	Size (nm)
①	50–300	Kp = 3.622, Ki = 74.2724, Kd = 74.2724	0.702	237	96.42
②	50–300	Kp = 3.622, Ki = 74.2724, Kd = 0.018846	0.345	234	90.9
③	30–300	Kp = 0.5, Ki = 0.05, Kd = 0.05	0.321	237	126.7

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
