# Peer review of "Development of Proportional–Integrative–Derivative (PID) Optimized for the MicroElectric Discharge Machine Fabrication of Nano-Bismuth Colloid"

_micromachines, 2020, doi:10.3390/mi11121065_

Round 1
Reviewer 1 Report
Dear authors,
You have done a good work.
Best regards,
Author Response
Journal:
Micromachines
Manuscript:
Original title: Parameter Control in Fabrication of Nano-Bismuth Colloids by Micro-Electric Discharge Machine
Changed title: Development of proportional-integrative-derivative (PID) optimized for the micro-Electric Discharge Machine fabrication of nano-bismuth colloid
Manuscript ID:
micromachines-1011510
Dear Reviewer,
Thank you for all the help and the useful comments of yours. We are really appreciated it. Thank you.
Sincerely,
Dr. Kuo-Hsiung Tseng
Professor,
Department of Electrical Engineering,
National Taipei University of Technology

Reviewer 2 Report
The paper has been hugely revised and improved. Nonetheless, I suggest to reorganize the information provided in the abstract, since, by reading the manuscript, it is quite clear that the main goal is the development of the PID and its tuning rather than the fabrication of nano-bismuth colloids.
It seems that the nanoparticle production is just an application of the developed control method for the proposed micro-EDM set up.
Therfore, I also recommend to modify the title of the manuscript, accordingly such as “Development of proportional-integrative-derivative (PID) optimized for the micro-Electro discharge machining fabrication of nano-bismuth colloid” or likewise. Conclusion also must be modified considering the aforementioned observations.
Minor modifications:
Abstract: line 10: “prepared by using…”
Intro: line 57 “current flows between the two electrodes”
Quality of Fig. 1 must be improved.
Section 2:
Regarding micro-EDM process parameters, I suggest to include a table reporting the values of the main process parameters (V, I, F, duty cycle, gain…etc)
Section 2.3. Line 266-269: the sentence “the higher the zeta…” until “stability” can be deleted. Line 276: please define di acronym of TEM (transmission electron microscopy) the first time it is recalled in the text. In the present version, the definition is at line 277.
Author Response
Dear Reviewer,
I am the corresponding author of the manuscript " Parameter Control in Fabrication of Nano-Bismuth Colloids by Micro-Electric Discharge Machine ". Thank you for all the help and the useful comments. We have revised the manuscript and detailed corrections are listed below point by point.
Q1 Therefore, I also recommend to modify the title of the manuscript, accordingly such as “Development of proportional-integrative-derivative (PID) optimized for the micro-Electric Discharge Machine fabrication of nano-bismuth colloid” or likewise.
- We have revised title of the manuscript. Please check our modified version.
Q2 Conclusion also must be modified considering the aforementioned observations.
- We have revised the conclusion in line 392 to 425 in the manuscript. Please check our modified version.
Q3 Abstract: line 10: “prepared by using…”
- We have revised the sentence line 14 with red text in the manuscript. Please check our modified version.
Q4 Intro: line 57 “current flows between the two electrodes”
- We have revised the sentence line 61 with red text in the manuscript. Please check our modified version.
Q5 Quality of Fig. 1 must be improved.
- We have reconstructed and revised the Figure 1 in the manuscript. Please check our modified version.
Q6 Regarding micro-EDM process parameters, I suggest to include a table reporting the values of the main process parameters (V, I, F, duty cycle, gain…etc)
- We have added preparation parameters line 293 to 297 in the manuscript (Table 1). Please check our modified version.
Q7 Section 2.3. Line 266-269: the sentence “the higher the zeta…” until “stability” can be deleted.
- We have removed the sentence line 272 to 274 in the manuscript. Please check our modified version.
Q8 Line 276: please define di acronym of TEM (transmission electron microscopy) the first time it is recalled in the text. In the present version, the definition is at line 277.
- We have revised the acronym of TEM in first time at line 268. Please check our modified version.
Thank you again for the useful comments. We look forward to the positive response.
Sincerely,
Dr. Kuo-Hsiung Tseng
Professor,
Department of Electrical Engineering,
National Taipei University of Technology

Reviewer 3 Report
The language throughout the paper has been improved and now I could better understand the content. Unfortunately, my worries about the automatic control application are still valid.
The paper is seriously flawed concerning the closed-loop control theory and application.
All the comments (besides English usage) are still valid and it seems that the writing about about the PID control and tuning is seriously flawed.
Besides all of the previous comments, I would add the additional ones, some according to the replies. Therefore, I will start with comment 8 (note again, no questions have been answered appropriately!).
Comment 9 (related to comment 5): Ki and Kd have units and they are not the same! Unit of Kd is time [e.g. second (s)], and unit of Ki is 1/time [e.g. 1/s]. Kp does not have unit. The mentioned sentences hold if the controller output and the controller input signals have the same physical units. If not, all the Kp, Kd and Ki units should be multiplied by additional units. In any case, Ki and Kd have still different units. Period. Equating Ki and Kd has no sense at all. Such attempts should be stopped immediately. If it works better, it still does not mean that it could work much better if they are not having the same value. You should search for the most appropriate values and not just, like in some recipe, “equating the values”. Equating the values only means that the authors have no idea what are they doing.
Coment 4: The windup phenomenon is cured by using anti-windup solution and not to change the controller parameters. The controller can always hit the limitations if close to the limits, no matter what controller parameters are used. Moreover, by increasing the Kd, the limitations are met more frequently when changing the reference! The answer provided in the reply indicate again that the authors have no idea what are they talking about.
Comment 6: If the simulation shows that the rise time is 0.001 ms it clearly means that such controller parameters are completely not useful in practice, since they can’t be in no sense achieved by the motor. Therefore, such solution should be immediately discarded and the new, better and slower solution should be found. As already told before, all of this makes no sense at all.
Comment 8 (related to Comment 2 and comment 3): In order to control the distance between electrodes the controlled value is the voltage. However, the paper is still very unclear what is the actual case. If the simulation response is much faster than the actual response, something should be done, for example decrease the speed of the closed-loop response toward something similar to the one which can be obtained in practice. Maybe even not using the PID control at all but some other kind of control algorithm, e.g. three-state controller.
To sum up, from the system control viewpoint, the paper seems to be seriously flawed (in several areas).
Author Response
Journal:
Micromachines
Manuscript:
Original title: Parameter Control in Fabrication of Nano-Bismuth Colloids by Micro-Electric Discharge Machine
Changed title: Development of proportional-integrative-derivative (PID) optimized for the micro-Electric Discharge Machine fabrication of nano-bismuth colloid
Manuscript ID:
micromachines-1011510
Dear Reviewer,
I am the corresponding author of the manuscript " Parameter Control in Fabrication of Nano-Bismuth Colloids by Micro-Electric Discharge Machine ". Thank you for all the help and the useful comments. We have revised the manuscript and detailed corrections are listed below point by point.
Q1 Comment 9 (related to comment 5): Ki and Kd have units and they are not the same! Unit of Kd is time [e.g. second (s)], and unit of Ki is 1/time [e.g. 1/s]. Kp does not have unit. The mentioned sentences hold if the controller output and the controller input signals have the same physical units. If not, all the Kp, Kd and Ki units should be multiplied by additional units. In any case, Ki and Kd have still different units. Period. Equating Ki and Kd has no sense at all. Such attempts should be stopped immediately. If it works better, it still does not mean that it could work much better if they are not having the same value. You should search for the most appropriate values and not just, like in some recipe, “equating the values”. Equating the values only means that the authors have no idea what are they doing.
- We thanks to reviewer for the correction of the unit of PID. In this study, we focus on fabrication nano-bi using micro-EDM. The details of our process nano-bi colloid:
- Create model of micro-EDM.
- Finding Kp, Ki, Kd use PID Tuner.
- Simulate the Kp, Ki, Kd values with MATLAB.
- Because the Ki value is greater than the VisSim limit, micro-EDM experiences an integral saturation effect.
- According to [Ang, K.H. et.al., PID control system analysis, design, and technology, 2005], to minimize the integral saturation effect by increasing the value of Kd.
- After conducting several changes in Kd's value, we got the optimal discharge rate when Kd equals Ki.
- Applying Kp, Ki, Kd to micro-EDM.
- The discharge rate increased compared to the manual process.
- After measuring the compliance of nano feature, we have succeeded fabricate the nano-bi colloid.
Q2 Comment 4: The windup phenomenon is cured by using anti-windup solution and not to change the controller parameters. The controller can always hit the limitations if close to the limits, no matter what controller parameters are used. Moreover, by increasing the Kd, the limitations are met more frequently when changing the reference! The answer provided in the reply indicate again that the authors have no idea what are they talking about.
- We agree with the reviewer that to cure the windup phenomenon is cured by anti-windup. However, our method has proven in our experiment. Based on [Ang, K.H. et.al., PID control system analysis, design, and technology, 2005] we can minimize the saturation integral effect by increasing value of Kd. In our experiment we change several times the value of Kd until we got the optimal discharge rate when Kd equals Ki.
Q3 Comment 6: If the simulation shows that the rise time is 0.001 ms it clearly means that such controller parameters are completely not useful in practice, since they can’t be in no sense achieved by the motor. Therefore, such solution should be immediately discarded and the new, better and slower solution should be found. As already told before, all of this makes no sense at all.
- Rise time and settle time are result of the simulation. In this study after we got value of PID, we clarify the value by plotting it in MATLAB (Figure 9). It confirms that group 2 has better response and micro-EDM able to fabricate nano-bi colloid.
Q4 Comment 8 (related to Comment 2 and comment 3): In order to control the distance between electrodes the controlled value is the voltage. However, the paper is still very unclear what is the actual case. If the simulation response is much faster than the actual response, something should be done, for example decrease the speed of the closed-loop response toward something similar to the one which can be obtained in practice. Maybe even not using the PID control at all but some other kind of control algorithm, e.g. three-state controller.
- The details of micro-EDM control the distance of the gap between the electrodes:
- We have the details of gap controlling in micro-EDM at line 221 to 232 in the manuscript. Please check our modified version.
- We thank to the reviewer, in future work we will use three-state controller to guide the electrode gap in micro-EDM.
Thank you again for the useful comments. We look forward to the positive response.
Sincerely,
Dr. Kuo-Hsiung Tseng
Professor,
Department of Electrical Engineering,
National Taipei University of Technology

This manuscript is a resubmission of an earlier submission. The following is a list of the peer review reports and author responses from that submission.
Round 1
Reviewer 1 Report
- The paper needs thorough English proofreading. Many sentences can’t be understood. If the revised version won’t be significantly improved in this manner, I would not be able to give any positive review.
Probably because of the problematic language, I was not able to understand the paper properly.
- Namely, it was not clearly explained what is the purpose of the PID controller. It seems that the PID controller controls the position of the motor connected to electrode. However, there is no explanation where is the relation between Figure 9 and 10. Where is the process output (the position of electrode) in Fig. 10?
- After once the most appropriate electrode distance is acquired, should it remain the same or must rapidly change during discharge process? Should it only change from “batch to batch” basis? It is not well explained in the paper.
Now, the most important comments:
- “Serious integral saturation effects” were reported on few instances in the paper. The authors should apply anti-windup protection instead. Please, study the solutions carefully and apply it on your system. Due to simplicity, the Conditioning technique should be a good starting point for anti-windup on PID controllers.
- In line 294 it was mentioned “if the value Ki equals Kd…”. It should be known that those two parameters cannot be compared since they even have different units. It would be similar to say: “when the voltage on a device equals the current”. It has no sense. Therefore in line 314 Kd should not be made equal to Ki by default. In no circumstances. First, the proper anti-windup should be applied, then the tuning should be repeated.
- In line 321 it is written “rise time is 0.001 ms, Settle Time is 0.004 ms”. You cannot have such rise or settle time with sampling time equal to 1 ms. Without any exception. Can you achieve such fast motor-driven movements in practice?
- Where is the figure of open-loop or closed-loop step-response during manual tuning? Open-loop should have an integrating character, but still would be beneficial.
Reviewer 2 Report
Below are my comments. I recommend major corrections of the “analysis of the nano-Bi” section.
The UV-vis spectra well-correspond to Bismuth nanoparticles, but the authors should be cautious when relating the size in table 1 to the wave numbers of absorption peaks.
There are no details about how the wave numbers were determined, but a difference of 3 cm-1 is to my opinion not realistic regarding the absorption curves in Fig. 11.
Furthermore, it is specified in the methods section that the size of the nanoparticles was measured by laser light scattering. However, no details are given about which apparatus were employed and in which conditions. The authors gave very precise values like 96.42 nm. So far as I know, the light scattering methods are far to be as accurate. Moreover, the size distribution of these kinds of nanoparticles is generally not monomodal. Instead giving a unique value, the authors should give the size distribution (in intensity) or at least, a deviation.
In the methods section, the authors cited zeta-potential measurements. But no results of zeta-potential were given then.
Looking at the TEM results, it is not easy to accept that individual Bi nanoparticles are formed. Instead, the TEM image suggests 400 nm aggregates of nanometric particles, joined with dense particles of hundreds nm, thus a fairly heterogeneous system. It may be due to the sample preparation for the TEM observations, unfortunately, no experimental details are given about this. Besides, how the authors may explain the difference between the size given by TEM and by light scattering?
Finally, an EDX analysis of the dense black particle would have been reliable, to ensure it was mainly Bi as well.
To conclude, the paper should be published thanks to corrections according to the comments below, but mainly with changes in the conclusion: (i) zetasizer must be replaced by dynamic light scattering. (ii) be less confident about the preparation of nano-grade Bi.

Reviewer 3 Report
The title of the present paper suggested a very interesting and appealing application of EDM technology and it also hinted an interesting strategy to produce Bi nanocolloid suspension implemented by EDM.
However, the paper has critical flaws about technical English language and grammar, and this fact prevented the present reviewer to read it properly and catch the real technical soundness of the method, procedure and strategy used and consequent evaluation of results.
All sections (Asbtract included) must be completely rewritten, as sentences often lack verbs and/or proper verb conjugations (at least) and punctuation is often incorrect.
It is very hard to read such a paper, as in the present form, the manuscript seemed to be written by non-experts of the field. Therefore, the paper cannot be accepted for publication. Before resubmitting the paper to the attention of this Journal, I warmly suggest the authors to operate heavy English language revision and ask for help in writing.
Reviewer 4 Report
Dear Authors,
Good job.
You have produced nano particles of Bismuth with electric discharge, and the attempted was achieved. I think you must characterize the particle composition due to the migration of oxygen and hydrogen from medium to the particles.
I, also do not understand How do you get a decrease in the voltage of x20? Which is the degree of ionization that you have in the medium?
Best regards,